# *Enteroaggregative E. coli* Adherence to Human Heparan Sulfate Proteoglycans Drives Segment and Host Specific Responses to Infection

**Anubama Rajan**[1], **Matthew J. Robertson**[2], **Hannah E. Carter**[1], **Nina M. Poole**[1], **Justin R. Clark**[1], **Sabrina I. Green**[1], **Zachary K. Criss**[3], **Boyang Zhao**[4], **Umesh Karandikar**[1], **Yikun Xing**[1], **Mar Margalef-Català**[5], **Nikhil Jain**[4], **Reid L. Wilson**[6], **Fan Bai**[7], **Joseph M. Hyser**[1], **Joseph Petrosino**[1], **Noah F. Shroyer**[3], **Sarah E. Blutt**[1], **Cristian Coarfa**[2,8], **Xuezheng Song**[7], **BV Venkataram Prasad**[4], **Manuel R. Amieva**[5], **Jane Grande-Allen**[6], **Mary K. Estes**[1], **Pablo C. Okhuysen**[9], **Anthony W. Maresso**[1]*

1 Department of Molecular Virology and Microbiology, Baylor College of Medicine, Houston, TX, United States of America, 2 Molecular and Cell Biology-Mol. Regulation, Baylor College of Medicine, Houston, TX, United States of America, 3 Department of Medicine Section of Gastroenterology and Hepatology, Baylor College of Medicine, Houston, TX, United States of America, 4 Verna and Marrs McLean Department of Biochemistry and Molecular Biology, Baylor College of Medicine, Houston, TX, United States of America, 5 Department of Pediatrics, Division of Infectious Diseases, Stanford University, Stanford, CA, United States of America, 6 Department of Bioengineering, Rice University, Houston, TX, United States of America, 7 Department of Biochemistry, Emory Comprehensive Glycomics Core, Emory University School of Medicine, Atlanta, GA, United States of America, 8 Dan L Duncan Comprehensive Cancer Center, Baylor College of Medicine, Houston, TX, United States of America, 9 Department of Infectious Diseases, The University of Texas MD Anderson Cancer Center, Houston, TX, United States of America

* maresso@bcm.edu

**Data Availability Statement:** All relevant data are within the manuscript and its Supporting Information files.

## Abstract

*Enteroaggregative Escherichia coli* (EAEC) is a significant cause of acute and chronic diarrhea, foodborne outbreaks, infections of the immunocompromised, and growth stunting in children in developing nations. There is no vaccine and resistance to antibiotics is rising. Unlike related *E. coli* pathotypes that are often associated with acute bouts of infection, EAEC is associated with persistent diarrhea and subclinical long-term colonization. Several secreted virulence factors have been associated with EAEC pathogenesis and linked to disease in humans, less certain are the molecular drivers of adherence to the intestinal mucosa. We previously established human intestinal enteroids (HIEs) as a model system to study host-EAEC interactions and aggregative adherence fimbriae A (AafA) as a major driver of EAEC adherence to HIEs. Here, we report a large-scale assessment of the host response to EAEC adherence from all four segments of the intestine across at least three donor lines for five *E. coli* pathotypes. The data demonstrate that the host response in the duodenum is driven largely by the infecting pathotype, whereas the response in the colon diverges in a patient-specific manner. Major pathways altered in gene expression in each of the four enteroid segments differed dramatically, with responses observed for inflammation, apoptosis and an overwhelming response to different mucin genes. In particular, EAEC both associated with large mucus droplets and specific mucins at the epithelial surface, binding

**Funding:** These studies were supported in part by grant U19 AI11497 that is as part of the U19 program (Novel Alternative Models of Enteric Diseases – NAMSED) from the National Institutes of Health assigned to MKE and AWM and from seed funds NAMSED institutional 1383006110 to AWM from Baylor College of Medicine. This study was partially supported by NIH P30 shared resource grant CA125123, NIEHS grants 1P30ES030285 and 1P42ES0327725 (MJR,CC) and by The Cancer Prevention Institute of Texas (CPRIT) RP170005. The funders had no role in study design, data collection and analysis, decision to publish, or preparation of the manuscript.

**Competing interests:** The authors have declared that no competing interests exist.

that was ameliorated when mucins were removed, a process dependent on AafA. Pan-screening for glycans for binding to purified AafA identified the human ligand as heparan sulfate proteoglycans (HSPGs). Removal of HSPG abrogated EAEC association with HIEs. These results may mean that the human intestine responds remarkably different to distinct pathobionts that is dependent on the both the individual and intestinal segment in question, and uncover a major role for surface heparan sulfate proteoglycans as tropism-driving factor in adherence and/or colonization.

## Author summary

*E. coli* is a significant cause of worldwide diarrhea and systemic infection. We previously established human intestinal enteroids as a model system to study *E. coli*-host interactions. We report the first large-scale assessment of the host response to infection from all four segments of the intestine across different donor lines for five different pathotypes. Whereas the host response in the duodenum is driven by the infecting pathotype, the response in the colon diverges by donor. We observed a major upregulation of host mucins, particularly for EAEC, which led to the identification of heparan sulfate proteoglycans (HSPGs) as the receptor on human colonoids. Our research highlights a new role for HSPGs for EAEC infection.

## Introduction

Enteroaggregative *E. coli* (EAEC) is a heterogeneous group of enteric bacteria that is a major cause of acute and persistent diarrhea, illness and death among children in developing countries that was originally isolated from children in Peru [1]. EAEC is also a cause of sporadic diarrhea [2, 3] and a common cause of traveler's diarrhea [4]. The clinical symptoms of EAEC infection include watery diarrhea [5] or bloody diarrhea [6], sometimes fever and mucoid stools [5]. EAEC displays a characteristic adherence pattern upon attachment to intestinal epithelial cells [1, 7, 8] and induces intestinal inflammation [9]. Some of the pathogenesis of EAEC may be related to a number of secreted protein effectors or toxins that are commonly associated with this diarrheal pathotype, including the serine protease autotransporters of Enterobacteriaceae (SPATES), [10–12]: ShET1 [13], Pic [14, 15], Pet [16], heat-stable enterotoxin 1 (EAST1) [17], hemolysin E (Hly1) [18], and dispersin [19]. There is no vaccine for EAEC or related diarrheal *E. coli* in general, although studies have focused on secreted toxins from related pathotypes [4, 20, 21]. Antibiotic resistance in EAEC has been reported and may be growing with evidence of horizontal gene transfer with extra-intestinal pathogenic *E. coli* (a cause of soft-tissue and, urinary tract infections, and bacteremia) [22–24]. Of great concern is the acquisition of virulence factors such as Shiga toxin, an event that contributed to the widespread outbreak in Europe in 2011 that sickened greater than three-thousand people and killed about fifty [25–27].

One of the more pressing biological and medical questions for diarrhea-causing *E. coli*, especially EAEC, is whether the variation in disease presentation (including the range of bacterial burden [28], length of colonization, and acute duration of illness [6, 29, 30]) is the result of host or bacterial genetics [31, 32], and whether the tropism of the organism amongst the four distinct segments of the intestinal tract is a driver of disease variability [33–35]. In fact, understanding the drivers of disease presentation is of central importance to probably all infectious

diseases, and more specifically with the examination of the human microbiota and its effect on human diseases. Such a point is emphasized in the concept of biogeography, which seeks to determine the relationship between locational colonization [34] or positioning, genetics, and downstream disease [32]. Inherent in the assessment of this question is the need to first identify where bacteria bind and establish an ecological niche; second, what are the consequences of this binding in terms of the host response, and third what are the bacterial and host molecular mechanisms that intersect during this response. Most knowledge of EAEC and related intestinal pathogens and their interactions with the host are obtained from transformed cell lines, ex-vivo culture techniques, or animal models. These resources have provided a wealth of information about the molecular pathogenesis of these and related pathotypes but often present limitations in recapitulating the physiology observed in humans [28, 36–38]. We recently reported the use of organotypic model systems such as human intestinal enteroids as a way to understand the molecular factors that drive the association of *E. coli* with the mucosal surface [35]. In this report, we substantially expand this analysis to include the concomitant host response to adherence for all four segments across three separate patient lines for various well-known *E. coli* pathotypes. Our results provide strong evidence for a patient and segment specific response to infection largely driven by an interaction with human heparan sulfated proteoglycans.

## Results

### Global host response to pathogenic E. coli

Previous studies from our group indicated that enteroaggregative *E. coli* formed unique adherence patterns to human intestinal enteroids that were dependent on intestinal segment and patient donor [35]. This adherence was also correlated to the presence of aggregative adherence fimbriae A gene (*aafA*) in infecting strains. Hypothesizing that the close association of *E. coli* in these aggregative morphotypes within the intestinal mucosa might correlate with the host response to the infection, we performed a large-scale transcriptome analysis of human intestinal enteroids across all four segments of the intestine (duodenum, jejunum, ileum, and colon) between at least three different patient donors with five different pathotypes of *E. col* (Fig 1A shows the general experimental scheme). In this regard, the duodenum, ileum and colon came from the same patient therefore allowing us to assess the effect of segment on the response, as well as compare the same segment between two different patients. In addition, we assessed responses to five different *E. coli* pathotypes. Each pathotype either causes a unique disease in the intestine or extra-intestinally. Such an approach allowed us to also determine how bacterial genetics within the same infecting species overlays on the other variables described above. The *E. coli* chosen for the study include a non-pathogenic control strain (*E. coli*: HS), enteroaggregative *E. coli* strain 042 (EAEC: a commonly used prototype strain that caused disease in a volunteer challenge study) [39], EAEC 042:ΔaafA (mutant strain lacking the major fimbrial gene a*afA*), enterohemorrhagic *E. coli* strain 8624 (EHEC prototype strain O157:H7, a Shiga toxin producing strain responsible for a food-borne disease in Oregon and Michigan [40], and extra-intestinal pathogenic *E. coli* (ExPEC, prototype strain CP9 that does not cause intestinal disease but is a major cause of bloodstream and urinary tract infections [41, 42]. These strains allow us to test the dependence of the host response on adherence (EAEC wild-type versus the *aafA* mutant), two related diarrheagenic strains with different disease pathologies (EAEC versus EHEC), intestinal infecting strains (EAEC and EHEC) versus an extraintestinal strain (ExPEC), and all pathogenic strains versus the non-pathogenic control (*E. coli* HS). A principal component analysis is an accepted approach to highlight global trends in large datasets and was performed here to examine the relationship of segment, patient, and

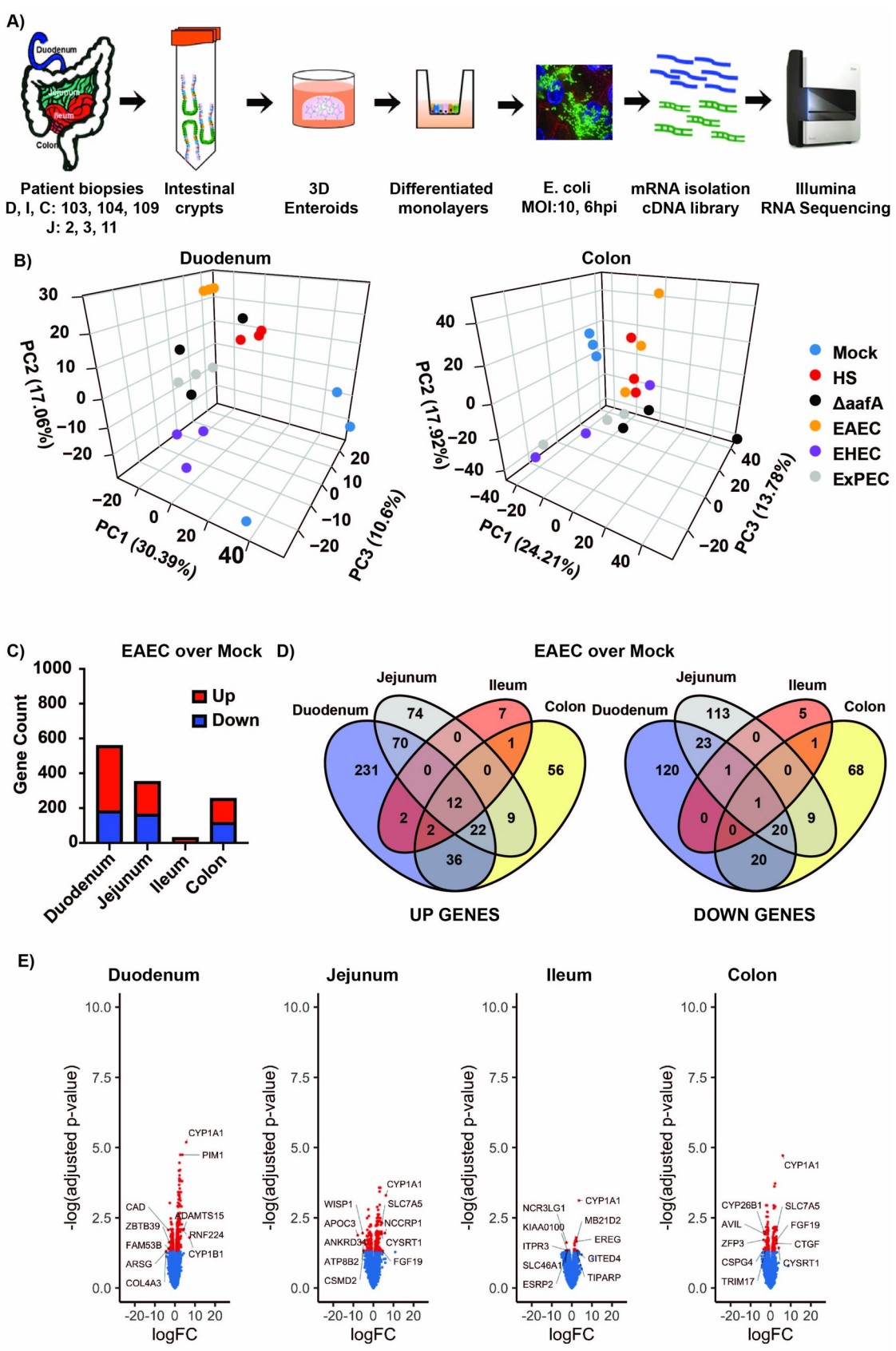

**Fig 1. RNA sequencing analysis of pathogenic *E. coli* infection of human intestinal enteroid monolayers (HIEMs).** (A) Experimental design and schematic representation for these experiments. (B) Principal component analysis of duodenum and colon monolayers demonstrating the variability observed in the samples (infected and mock controls). (C) Total number of significant genes identified for the EAEC infected group (BH adjusted p-value < 0.05). (D) Venn diagrams present the number of genes that are differentially expressed within the duodenum, jejunum, ileum and colon of EAEC infected samples compared to the uninfected control (E). Volcano plots of differential gene expression in each tissue for the EAEC infected samples compared to the mock infected controls. Red dots indicate genes that have an adjusted p-value of at least 0.05 with a linear fold change of at least 1.5.

strain to each other (Fig 1B). Interestingly, there were significant differences in the response between the most distal and proximal intestinal segments. In the duodenum, there was substantial variance in the baseline host response in uninfected samples between patients described by PC3 (Fig 1B, left panel, blue dots) that converged on a similar host response when bacteria, regardless of the pathotype or strain, were added (Fig 1B, left panel, all dots) described by PC1. In the duodenum from the PCA analysis the infection event drives the differential gene expression regardless of bacterial phenotype, with the bacterial phenotype, and patient identity, contributing to a lesser extent. The colonoids exhibited a less orchestrated response. The mock infected samples for three different patients showed a highly similar host response (Fig 1B, right panel, blue dots) that became highly scattered once infected, regardless of strain (Fig 1B, right panel, all dots). These associations were statistically significant (p < 0.05). This suggests that whereas the duodenum had a more similar response between patients when *E. coli* of any type were used, the colon has responses that were highly patient specific and varied dependent on the infecting organism. These trends were generally true for tissues that bordered these segments (i.e. the jejunum resembling more the duodenum and the ileum resembling the colon–S1A Fig).

We next assessed the general host response to infection, focusing mostly on EAEC. The duodenum, jejunum and colon demonstrated significant changes in gene expression, the duodenum having the highest of the three, and with the number of upregulated genes three times higher than the number of downregulated genes (Fig 1C). Remarkably, the ileum had little change in gene expression, either up or down, a trend observed for all *E. coli* pathotypes or strains assessed (S1B Fig, S1 and S2 Appendices). Similar to EAEC response, the duodenum showed the greatest number of gene expression changes, with EHEC and ExPEC leading the way. Of note, EAEC induced the least number of gene changes among all the strains that were assessed, even less than the non-pathogenic control. Interestingly, the EAEC lacking the fimbrial gene *aafA* had over three times and over seven times the number of total gene expression changes in the duodenum and colon, respectively, than the wild-type strain (discussed below). A significant segment-dependent change in gene expression was observed with only 13 genes shared between the four segments (Fig 1D). For example, there were 306 different genes downregulated in each individual segment that were not shared between segments compared to just 75 shared by at least two segments (368 vs 152 for upregulated genes). Of over 13,000 profiled host transcripts, 560 were significantly altered in duodenum, 354 in jejunum, 32 in ileum and 257 in colon during EAEC infection (Fig 1E, S4 Fig and S2 Appendix), less than 10% of them overlapping between segments. Volcano plots are useful because they can show the dynamic range of the level of responses in the data set. As shown in Fig 1E, upregulated genes had a much larger range of expression than did the downregulated genes, with the duodenum have the greatest spread of all four segments and the ileum the least, a trend that was generally true for all *E. coli* pathotypes or strains (S2 Fig, S1 and S2 Appendices).

## Gene expression signatures in segments and patients for E. coli pathotype infections

Hierarchical clustering can be used to determine how "related" gene responses between segments and pathotypes are to each other. This allows one to ask questions as to whether certain

intestinal regions or pathogens behave like others, and so on. We used Pearson correlation coefficient analysis to examine the relationship, displayed in a dendrogram format, between various infection groups. As shown in Fig 2A, and consistent with data represented in Fig 1, the responses in the duodenum were highly consistent within each pathotype or strain between each of the three different patients examined. This means that in the duodenum, the response is less driven by the host and more by the infecting organism. In this regard, all the bacterial infections (HS, EAEC, EHEC, and ExPEC) were more similar to each other than mock, and, rather surprisingly, EAEC was more similar to HS than any others (including a strain with an identical background but lacking the fimbrial gene a*afA*–Fig 2A). This is in complete contrast with the responses observed in the colon; only the mock-infected colonoids were most related to each other amongst the three different patients. In fact, the responses were so divergent between the patients that there was no observable pattern that was significant, suggesting that the individual patient response to the organism was more important than the organism itself. When the breadth of up and down regulated genes are clustered into a heat map and compared between each bacterium across each segment, EAEC demonstrated the fewest changes amongst all assayed genes while EAEC lacking the fimbrial *aafA* gene demonstrated the most (Fig 2B). As was observed for EAEC, there were far more unique genes expressed in each individual segment than there were shared between segments for each of the *E. coli* pathotypes or strains, thus signifying the truly unique response of each of these tissues to these bacteria (S3 and S4 Figs). The notable exception to this was again EAEC lacking the fimbrial gene which produced the highest number of total up- and down-regulated genes shared in common between all four segments. A Gene Set Enrichment Analysis (GSEA) was performed on the differential expression gene sets for each infection. The most common pathways that were enriched are mostly involved in adaptive immune response, cellular differentiation, metabolism, phagosome activation, G-Protein Coupled Receptors, G-alpha signaling and mitosis (S5 Fig and S3 Appendix). One of the more striking observations was changes in apoptosis and/or cell death responses; every pathotype, and especially the EAEC lacking *aafA*, had a positive upregulation of cell death pathways. We confirmed that EAEC was directly altering cell responses by examining the thickness of the intestinal enteroid epithelium during a six-hour infection using Periodic acid-Schiff Stain/Alcian Blue (PAS-AB). Whereas the mock infected samples maintained an epithelial layer of close to 80 microns (Fig 2D), upon infection with EAEC this thickness decreased on average 4-fold, which we interpreted as caused mostly by Goblet cells releasing their mucus (Fig 2E), indicating that the epithelium was responding by attempting to increase barrier integrity. Note that no cell rounding, monolayer sloughing, or reduction of TER was observed at a similar time point for these infections [35]. The mechanistic underpinnings of these observations are currently unknown but might relate to the findings discussed below.

One of the groups of genes that had a significant up-regulation in expression across most of its paralog members were mucins. Mucins are apically localized proteins that often are glycosylated at one or more sites along the polypeptide chain. In the intestine, they are a major component of the mucus layer [43]. We assembled 10 individual genes encoding different mucins for which the number of reads in our gene expression analysis was above the statistical threshold for stringency and plotted their expression as a heat map across all four segments and four pathotypes (including the EAEC fimbrial mutant–Fig 2C). Remarkably, in 31 of the 40 (77%) possible scenarios across all four segments, EAEC infection led to an up-regulation of one or more mucin genes over the mock-infected control. In general, infection with any bacteria of any type overwhelmingly followed the same trend with the greatest responses being observed in the duodenum and jejunum (Fig 2C). Muc 1 and Muc 13 were upregulated under every condition (40/40). Nine of the ten genes were universally upregulated in the duodenum and

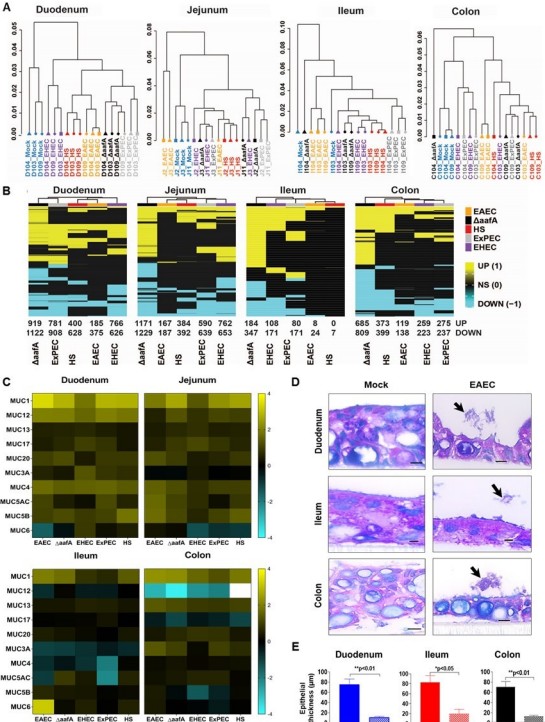

**Fig 2. RNA-Sequencing analysis reveals a pathogen-specific differential gene expression signature within different segments of intestine.** (A) Hierarchical clustering dendrogram of all gene expression changes using the Pearson correlation coefficient among mock and *E. coli* infected samples for all four segments of intestine (Duodenum, Ileum, Colon from patients 103, 104 & 109 and jejunum from patient 2, 3 & 11 –see Methods) (B) Samples were hierarchically clustered based on the union of differentially expressed genes across all comparisons (false discovery rate (FDR) < 0.05 and fold change (FC) exceeding 1.5x). Genes were converted to 1 for up-regulated genes and -1 for down-regulated genes and 0 if it a genes were not significant in a particular signature. Samples were clustered used the Euclidean distance. (C) Heatmap showing EAEC infection-specific regulation of *Muc* gene expression in HIEs. (D) PAS/AB staining of HIEMs infected with EAEC shows epithelial cell damage. The left panel shows the mock-infected epithelial cell layer whereas the right panel reveals epithelial thinning and damage caused by EAEC adherence (EAEC clusters denoted by arrows). (E) Quantification of epithelial thinning caused by EAEC infection. Scale bars indicate 10μm. Epithelial thickness was quantified from average of 12 images taken at 100x and with field size of 2560x1920 pixel. Unpaired t-test was carried out, error bars denote SEM and ** indicates significantly different (P<0.01), * indicates significantly different (P<0.05).

jejunum, signifying a strong general mucin positive response in these enteroid segments. Only one gene, that of Muc6, was either unchanged or downregulated under these conditions. Somewhat unexpectedly, and in contrast to the duodenum and jejunum, the ileum and colon, although displaying an overall up regulation of Muc genes, showed eight of the ten genes that had either no change or a down modulation of the gene, with Muc12 being either unchanged or upregulated in 20 of 20 cases. These results suggest the enteroids undergo a strong general up regulation of most mucins to the presence of bacteria, regardless of the pathotype, but that there are significant segment-specific responses, especially at the most distal and proximal ends of the alimentary canal, with some mucins seemingly being purposely down modulated.

## Adherence of EAEC to human intestinal enteroids

The finding that EAEC forms different adherence patterns on enteroids from each intestinal segment [35], the significant host response induced by EAEC in all four segments, and the generalized upregulation of many mucin genes in response to infection, prompted a closer

scrutiny of this unique host-pathogen interaction. We used scanning electron microscopy (SEM) to examine the interaction of EAEC with enteroids from the duodenum, ileum, and colon. As shown in Fig 3A–3C, mock-infected cells showed a normal undulating topography with occasional droplets of mucus (white balls marked with green asterisks). EAEC-infected cells formed biofilm-like aggregates on the duodenum and colon and classic aggregative adherence on the ileum, findings consistent with previous publications [35]. Overall, the surface of the infected cell looked very similar to mock-infected cells with the notable exception that EAEC seemed to preferentially cluster around the droplets of mucus (Fig 3A–3C, Row 2). In fact, when images were taken in areas of the mucosal epithelium that lacked these mucus droplets, no bacteria were observed (Fig 3A–3C), Row 3). Quantification of this observation indicated that there was an average increase of 5–10 times in the level of bacteria associated with mucus droplets (Fig 3D). When the experiment was repeated with EAEC lacking the *aafA* gene, very few bacterial cells were observed, and those that were observed, bound to the surface by atypical adherence (Fig 3A–3C, Row 4)). Transmission electron microscopy demonstrated that many EAEC preferentially and closely associated with Goblet cells (Fig 3E). Finally, the binding of individual bacterial cells or aggregates of bacteria to the enteroid surface resulted, with time, in the large outgrowth of aggregated bacteria that grew upwards from the point of origin (Supplemental Video File 1). This behavior was not observed for the control non-pathogenic strain HS or EAEC that lacked the a*afA*.

To further understand the nature of this association, we performed immunofluorescence microscopy of colonoids infected with EAEC expressing the green fluorescent protein while simultaneously staining for MUC2, a goblet cell secreted mucin (Fig 4). MUC2 was not one of the genes that met the threshold for reads in the gene expression analysis but is a mucin previously reported to be associated with other diarrheagenic pathotypes such as EHEC [44, 45]. As shown in Fig 4A, mock-infected cells were effectively stained with DAPI, MUC2, and E-cadherin allowing for easy visualization of cell borders and nuclei. Interestingly, although EAEC associated with MUC2 (red), there was a sizable fraction that did not. This is further appreciated using a Z-stack analysis (Fig 4B) whereby approximately one-half of EAEC clustered directly above MUC2 but the remainder seemed to intimately associate with the cell surface in areas that lacked MUC2 staining. To further investigate this phenomenon and gain a better structural appreciation of this observation, we used "apical-out" colonoids reported by Co et al. [46]. Apical-out enteroids have their apical surface facing the media while their basolateral surface is directed inwards, a convenient feature that allows for a 3-dimensional analysis of the bacterial association with colonoids. As observed in Fig 4C, whereas there was definite clustering of EAEC around clusters of MUC2, there was also a significant association of EAEC with the epithelial surface that lacked MUC2, especially pockets of bacteria that wedged themselves either between the cells or in the folds of the apical-out structure.

To determine the importance of mucus in contributing to the total level of EAEC binding to the surface of intestinal enteroids, we used two contrasting chemical-based approaches to either enhance (DAPT, N-[N-(3, 5-Difluorophenacetyl)-L-alanyl]-S-phenyl glycine t-butyl ester) or remove (NAC, N-acetyl glucosamine) the levels of mucus on the surface of colonoids (Fig 5A). As shown visually via Giemsa staining (Fig 5B), and quantified (Fig 5D), when the DAPT concentration was increased to 10 µM, so did the number of aggregated EAEC bound to the surface. In contrast, as the concentration of NAC was increased, a dose-dependent reduction in the number of aggregated EAEC on the enteroid surface was also observed (Fig 5C and 5E). When examined via the use of immunostaining for MUC2 as a proxy for the presence of mucus (Fig 5F), a marginal increase in MUC2 levels and fluorescent EAEC bound was observed after DAPT treatment whereas there was complete ablation of MUC2 and very little EAEC bound when enteroids were pre-treated with NAC, findings generally consistent with

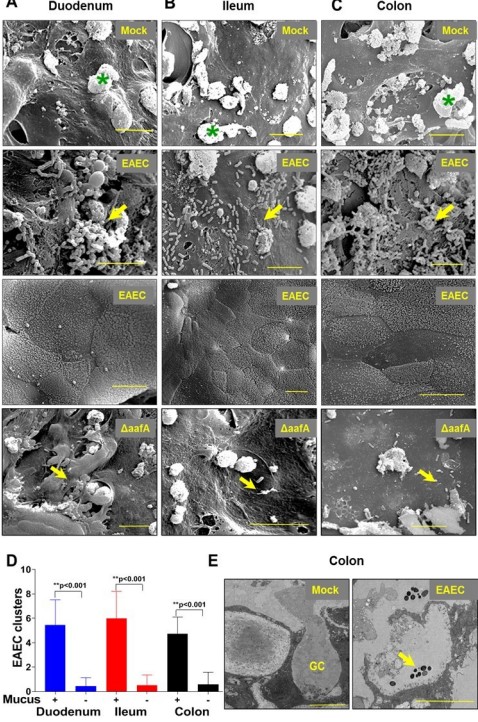

**Fig 3. SEM and TEM images of EAEC adherence to human intestinal enteroid monolayers.** (A-C) shows SEM image of Mock, EAEC and EAEC: Δ*aafA* infected duodenal, ileal and colon differentiated HIE monolayers. Row one shows differentiated enteroid monolayers covered by secreted mucus droplets (denoted by the asterisk). Row two shows adherence of EAEC to mucus (yellow arrow) present on the duodenal, ileal and colon monolayers. Row three shows EAEC infected duodenal, ileal and colon monolayers but with no EAEC adherence and without the presence of mucus droplets. Row four shows EAEC: Δ*aafA* infected duodenal, ileal and colon monolayers. (D) Quantification of the EAEC aggregates in the presence and absence of mucus droplets on enteroid monolayers was quantified from average of 12 images taken at 1700x and error bars denote SEM and *** indicates significantly different (P<0.001). (E) TEM of colonoid monolayers showing vesiculation of Goblet Cell (GC) mucus and EAEC adherence to GC mucus (yellow arrow). Scale bars indicate 10μm.

the Giemsa staining noted above. Thus, it seems as though a significant portion of EAEC binding to human colonoids is dependent on functional mucus (or surface carbohydrates that are sensitive to these agents), but that only some of that adherence can be directly attributed to MUC2.

## EAEC aggregation and association with human intestinal enteroids occurs through sulfated glycans

The finding that EAEC adherence was dependent on aggregative adherence fimbriae association with host mucus prompted a search for the glycan receptor that drives this process. We cloned DNA encoding amino acids 1–135 of aggregative adherence fimbriae A (AafA) separated by a 10-residue linker into pQE-30 and expressed the polypeptide as a hexa-his fusion protein, and purified AafA using Ni-His agarose affinity chromatography. As shown in S6A Fig, AafA purified largely as a single, full-length polypeptide that was greater than 95% pure as assessed by SDS-PAGE. To determine the glycan (s) that may interact with AafA, we screened the purified protein for binding to a synthetic glycan array comprising >500 glycans [47]. This microarray includes many synthetic glycan structures from glycoproteins and glycolipids, but not complex glycosaminoglycans (GAG). Interestingly, while no binding of AafA was

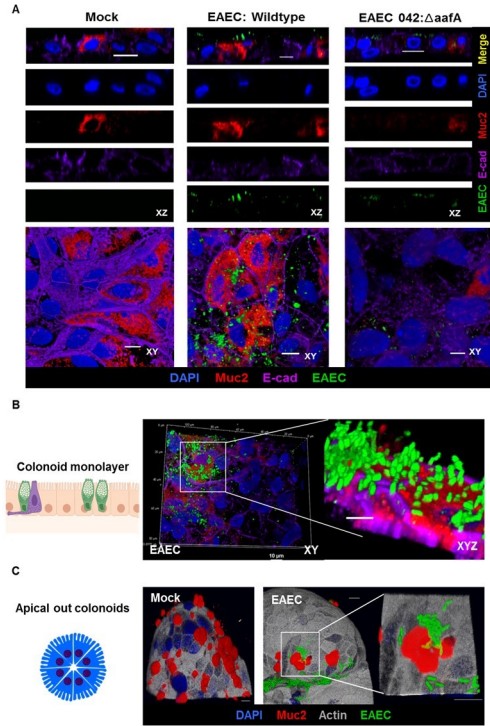

**Fig 4. Association of EAEC to intestinal mucus (MUC2) to TC202, colon line as revealed by immunofluorescence micrographs.** (A) Representative confocal micrographs of HIEMs stained for nuclei (DAPI, blue), MUC2 (Red), E-cadherin (Purple) and EAEC (GFP, green). HIEMs were either mock treated or infected with EAEC A2A:GFP or EAEC:ΔaafA at a MOI of 10 for three hours. (B) Volume projections of EAEC A2A:GFP infected HIEMs (C) EAEC infection on apical out colonoids. 3D reconstruction images show EAEC growing above goblet cells secreting MUC2. (Inset shows higher magnification). Scale bars indicate 10μm.

observed at low concentrations (S6B Fig), there was some binding events above what is considered background (< 2,000 relative fluorescence units (RFU), background RFU that contributes to non-specific binding). While no clear binding specificity towards certain structures was observed, AafA seemed to show binding preference towards "negatively charged" glycan structures in this analysis (S6B and S6C Fig). Thus, a second array consisting of charged glycosaminoglycans (GAG), including oligomers from hyaluronic acid, chondroitin sulfates and heparan, were tested for binding to AafA. Strikingly, AafA showed strong and specific binding (10–50 times more binding) for compounds that consisted of oligomers of heparan. Heparan is a highly sulfated GAG polysaccharide sharing basic structures with heparan sulfates, the GAG chain attached to heparan sulfated proteoglycans (HSPGs–Fig 6A). Fig 6B demonstrates the core structure of HSPGs consisting of Glucuronic/iduronic acid (GlcA) and N-acetylyglucosamine (GlcNAc). To determine if HSPGs could be suitable ligands for AafA, we modeled the heparan sulfate analog N, O6-DISULFO-GLUCOSAMINE (SGN) onto the published NMR structure of AafA (Fig 6C) using UCSF Chimera. Indeed, a PATCHDOCK analysis predicted a potential site for the interaction of heparan sulfate with AafA based on electrostatic interactions [48–50], mediated by the residues H62, N61 and Q64 (Fig 6C, inset). To determine if indeed HSPGs represented the receptor for EAEC's interaction with colonoids, we pretreated the monolayers for two hours with the enzyme heparinase. As shown in Fig 6D and 6E, whereas untreated colonoids retained the characteristic biofilm-like aggregation and binding of EAEC, treatment with heparinase substantially reduced association with EAEC. Similarly,

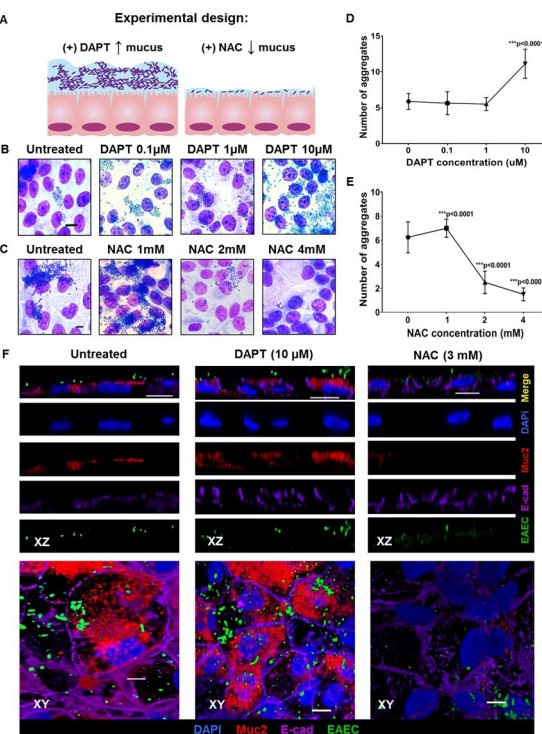

**Fig 5. : EAEC adherence is dependent upon host intestinal mucus.** (A) Experimental design explaining the chemical treatment performed to manipulate intestinal mucus and (B) showing increasing EAEC aggregative adherence to HIEMs with increased mucus production as induced by DAPT (a gamma secretase inhibitor). (C) EAEC clusters on HIEMs in the presence of N-Acetyl Cysteine (NAC), a mucolytic agent. (D-E) Quantification of EAEC aggregates in response to increasing doses of DAPT and NAC was quantified from average of 12 images taken at 100x and with field size of 2560x1920 pixel. Tukey's multiple comparison test was performed, error bars denote SED and *** indicates significantly different (P<0.001). (F) shows representative confocal micrographs of HIEMs stained with fluorescent indicators for nuclei (DAPI, blue), MUC2 (Red), E-cadherin (Purple) and EAEC (GFP, green). HIEMs were infected with EAEC A2A:GFP that were treated with DAPT (10 μM) and NAC (3 mM). Representative of over 50 images. Scale bars indicate 10μm.

adding soluble heparin to the infection completely abolished any association of EAEC with the colonoids (Fig 6F), presumably via competitive inhibition with HSPGs present on the HIE surface. The results were confirmed after being quantified by assessing the number of bacterial colony forming units (CFUs–S6D Fig) before and after treatment. No effect on bacterial viability was observed with this concentration of heparin. These data strongly suggest that EAEC interacts with human intestinal enteroids via the direct binding of aggregative adherence fimbriae to heparan sulfated proteoglycans.

## Discussion

Intestinal organoids or enteroids have been proposed as an alternative model system to study the molecular aspects of host-pathogen interactions, especially as they apply to the gastrointestinal tract [44–46, 51–57]. This approach is built on certain virtues that organotypic systems possess, including that they are untransformed and human, heterocellular [58–60], grow in three dimensions [46], ability to engineer "enhancements" such as co-culture with immune cells [61], flow in one direction [62], and vasculature [63] that undergo physiologic responses to stimuli such as fluid secretion [51, 64, 65], mucus production [45], and innate immune responses [56, 61]. In addition, researches could (i) assess all four intestinal segments from

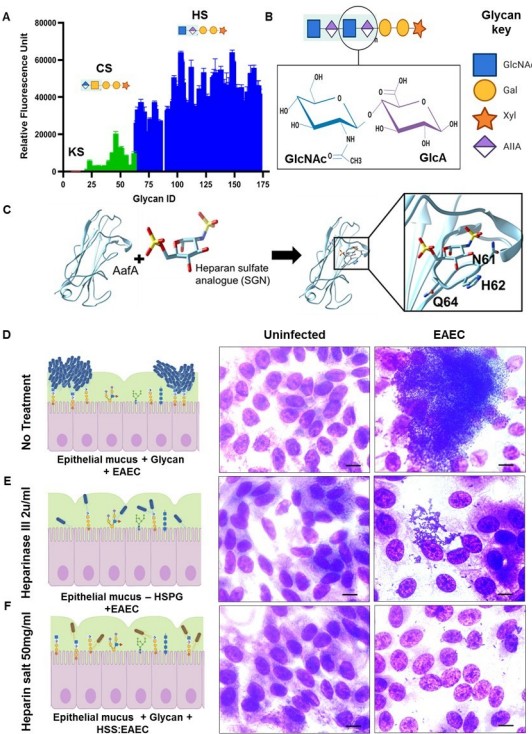

**Fig 6. HSPG mediates adherence of EAEC to HIEMs.** (A) GAG microarray of AafA shows high affinity binding to heparan sulfate GAGs. (B) shows the core chemical structure of HSPG. (C) Docking of SGN molecule on AafA reveals electrostatic interactions mediated by the residues H62, N61 and Q64. (D-F) Removal of HSPG by pretreatment of HIEs with heparinase III or addition of Heparan Sodium Salt (HSS) to EAEC as a competitive inhibitor exhibited reduced EAEC adherence on HIEs. Representative images, scale bars indicate 10μm.

both the same or different patients [35], (ii) generate a growing library of patient lines that allow one to examine the role of host genetics in pathogenic responses [51–55, 57, 66, 67], (iii) advance technological improvements for enhancing their growth, passage, storage [68], and reverse genetics [69, 70]. Organotypic models do have limitations, the most notable of which is that they cannot reproduce the full extent of mammalian disease. However, since the interaction of pathobionts and the native microflora with the host mucosa surface is likely the first step in colonization and/or pathogenesis, and organotypic cultures generate mucosal surfaces that resemble intact intestines with patient-specific responses, it would seem that such systems are excellent models to study the molecular aspects of these interactions. It stands to reason then that the approach presented here can be specifically adapted to ask pertinent questions about both the host response to diarrheagenic *E. coli* and the mechanisms by which these pathotypes bind or colonize.

In this regard, we conducted the first comprehensive evaluation of the host response of human intestinal enteroids from all four segments for three separate donors to a bacterial pathogen (here various *E. coli* pathotypes), and report the following original findings; (i) that the baseline duodenal response in the absence of infection was highly variable but converged in a pathogen specific-manner that was seemingly independent of donor. This is similar to the response of small intestinal jejunal cultures to the viral gastrointestinal pathogen human rotavirus [56] but it contrasted with the colon enteroids, whereby a highly similar baseline uninfected response underwent a significant divergence after infection that was substantially driven by the donor; (ii) that the total number of genes altered in response to any *E. coli*

infection generally followed the pattern duodenum > jejunum > colon > ileum, with the ileum only accounting for less than 5% of all gene changes; (iii) that EAEC produced the least amount of changes in overall gene expression of any of the four *E. coli* pathotypes. Paradoxically, EAEC lacking aggregative adherence fimbriae A yielded the highest number of genes that changed in expression among all *E. coli* tested. There are two interpretations of this data. First, it may be that the *aafA*-negative strain, being unable to adhere, upregulates the production of toxins or other effectors (possibly the SPATES) compared to wild-type, which in turn elicit a significant response from host. A second interpretation is that adherence actually pacifies the host, i.e. the wild-type strain clandestinely adheres without triggering many gene responses in the HIEs and that the *aafA* mutant strain loses this control via loss of adherence; (iv) that the gene expression responses in each intestinal segment were generally not shared between segments. In fact, for EAEC infected enteroids, only 12 genes (out of > 400) went up in expression, and only a single gene down (out of > 300) in *all* segments. This trend was generally true for all pathotypes; (v) there was a strong correlation between infection and the upregulation of genes involved in inflammatory pathways, apoptosis, or cell death responses. Interestingly, and rather unexpected, was a general upregulation in all segments of enteroids in the genes that encode mucins, although not universally shared between each segment. This was especially true for EAEC infection; (vi) that EAEC associated with enteroid mucus on the apical surface (which was dependent on AafA), especially MUC2-positive areas, but that there was substantial association with the cell surface that lacked Muc staining; and, finally, that (vii) this association with the enteroid surface occurs through the selective binding of heparan sulfated proteoglycans (HSPGs), which, the data suggests, is the receptor for AafA. The significance of two of these major findings, host response and adherence through HSPGs, are discussed in detail below.

It is increasingly being recognized that mucosal glycans, and mucus in general, are important determinants of colonization, tropism, adherence, and disease [43, 44, 71, 72]. Types and glycosylation pattern of mucins vary along the GI tract and can be modulated by host genetics, gut microbiota, diet and co-infections with other diarrheal pathogens [44, 73–78]. Specifically, the gastrointestinal tract (GIT) contains two different type of mucins: transmembrane mucins and gel-forming mucins, the latter secreted by goblet cells [77, 78]. Although the distribution of the mucus layer varies across the GIT, they are composed of water, ions, proteins, lipids, proteoglycans (PG) and glycosaminoglycans (GAG) [79–82]. GAGs are a linear polysaccharide and are classified into four major groups: heparan sulfate glycosaminoglycans (HSGAGs), chondroitin sulfate/dermatan sulfate glycosaminoglycans (CSGAGs), keratin sulfates and hyaluronic acids [83, 84]. GAGs show variation in their distribution in the GIT [82]. Clinical data indicates that EAEC either associates with, or may induce the production of, mucus. For example, mucoid stools are one clinical characteristic observed in patients infected with EAEC, including in outbreaks in young children or infants [85, 86]. EAEC secretes plasmid-encoded toxin (PET) that is associated with mucoid stools [16] and a protein involved in intestinal colonization (PIC) that has been characterized as a mucinase [14, 15]. The interaction with certain mucins such as Muc1 may also drive inflammatory responses [71] and in a murine host, EAEC induces the emptying of Goblet cell mucus, seems to associate with this mucus, and may even enter Goblet cell vacuoles [28]. In vivo organ culture analysis indicates EAEC embeds itself in a mucus layer [8]. Here, adhered EAEC associates with large droplets of mucus on the mucosal surface. Chemically induced stimulation of mucus production increased EAEC adherence whereas chemically induced removal of mucus eliminated most adhered bacteria (especially large clusters) while preserving small pockets of individually adhered cells. Specific staining of mucus for MUC2, one of the known dominant mucins enriched in the intestinal mucosa, showed clear association of fluorescent EAEC with MUC2

clusters, but also a substantial (estimated to be about half) of EAEC intimately associated with the host cell in MUC2-free areas. Strikingly, the deletion of aggregative adherence fimbriae A nearly abolished all association with mucus generally or MUC2 specifically, consistent with earlier studies from our group [35]. When taken together, this data strongly indicates that a major fraction of EAEC adherence, and probably by extension ability to colonize, is dependent on AafA to ligate host mucus for strong adherence to the mucosal surface. The necessary question then becomes; what is the host ligand that dictates this association and what are the downstream consequences of it?

We formed the hypothesis that EAEC association with the human intestinal mucosa might partially proceed through the selective binding of one or more types of carbohydrates. Since binding to human enteroids is strongly dependent on aggregative adherence fimbriae, we reasoned that this protein might potentially contain binding sites for human sugars. A large-scale screen examining the binding of AafA to over one-thousand unique imprinted sugars yielded heparan sulfated proteoglycans (HSPGs) as a strong candidate host receptor for EAEC. The strength of this claim is further backed by the following supporting evidence; (i) over 90% of all the glycans assayed for binding to AafA yielded binding that was at or below background levels, thus arguing HSPGs are specific for AafA; (ii) the specific removal of heparan sulfated sugars on the surface of colonoids using soluble heparinase abrogated binding of EAEC; (iii) the addition of excess soluble heparin sulfated sugars to the media during an EAEC infection also abrogated binding of EAEC; (iv) molecular docking experiments using the published NMR structure of AafA demonstrated that heparan sulfate analogs fit snugly into a putative ligand binding site making strong ionic contacts with H62, N61 and Q64; and finally, (v) these interactions resulted in large clusters of aggregated bacteria that grew in real-time from their point of origin, a process not observed for the control *E. coli* HS or EAEC lacking AafA. These findings suggest a model by which EAEC associates with the intestinal mucosa via AafA binding to sulfated heparan glycans, a process that leads to the aggregative phenotype, and likely colonization (S7 Fig).

HSPGs are unbranched negatively charged polysaccharides linked to extracellular matrix and epithelial cell surface proteins. At least ten different genes mediate these modifications, resulting in a wide array of different sulfation and glycosylation structures on the surface of cells [87]. Although the exact structure of the HSPG that binds AafA is not known, judging from the multitude of positive signals for the various HSPGs tested in Fig 6, AafA is likely to accommodate various terminal glycan structures similar to the core heparan sulfate structure in its binding pocket. Since the abundance and type of HSPG structure one displays on their intestinal epithelium may be influenced by the extensive array of genes that generate HSPGs, it is interesting to speculate that susceptibility to or severity of EAEC-induced diarrhea or colonization may be determined by the composition of one's HSPG surface structures. In addition, some of the native ligands of HSPGs include chemokines, cytokines, and growth factors leaving one to also wonder whether EAEC adherence may dampen the pro-inflammatory response one would mount to an infection. The finding that HSPGs are a ligand for EAEC are particularly striking in light of the fact that these carbohydrates are well known receptors for many human pathogenic viruses, including human papillomavirus, herpes simplex virus, and dengue virus, with another dozen viruses suspected as using HSPGs in some way for cell binding or entry [88]. To our knowledge, the only known intestinal bacterium for which there is a report of binding to HSPGs is *L. monocytogenes*, which was reported to bind syndecan-1, a glycosylated surface protein important in cell migration and proliferation [89, 90]. Syndecan-1 may represent one possible protein target receptor for EAEC and related *E. coli*. Nataro et al. reported that one potential ligand for EAEC might be extracellular matrix proteins such as fibronectin [91], an interaction mediated by AafA. Fibronectin associates with HSPGs via its

HS chains [92]. Our data suggest that EAEC may interact with ECM proteins via their association with HSPGs, possibly including fibronectin as a co-complex. Since many bacterial pathogens have been reported to interact with ECM proteins, it will now be necessary to determine if HSPGs are a component of these interactions [72, 93–95].

Analysis of the host enteroid response to infection produced some striking observations. The first is that the most significant divergence in host response from the baseline uninfected state to different pathotypes occurred in the colonoids. There was no uniform agreement in gene expression between any of the three separate donors, even to the same infecting organism. This data suggests that the colonic response to bacteria is highly variable and driven by the patient; a necessary extension of this finding is that the underlying genetics of the host are a major driver of the response. Since the level and type of binding can differ between colonoids from different patients, and binding is dependent on HSPGs, our data also suggests one should consider heterogeneity in epithelial glycosylation patterns as a player in such responses, and colonization by extension. These molecular findings align well with the multitude of different clinical manifestations sometimes observed with infection with diarrheagenic *E. coli*, especially EAEC. Second, of the 20 different infection groups assessed in this study, each with three patients per group, in every single case, the total number of differentially regulated genes in the duodenum was the greatest of all four segments. This compelling data suggests that we cannot ignore the proximal small intestine as a major player in the response to infection. It is tempting to think about the duodenum as a sort of first line responder to an ingested microorganism, this tissue being the first intestinal segment to encounter a bacterium being passed through the GIT. In this regard, we hypothesize that the duodenum may respond by increasing the number of pro-inflammatory chemokines that may serve to activate the host immune response in downstream segments. This is consistent with numerous cytokine and chemokine genes upregulated by all pathotypes in this system. Conversely, the ileum was the segment that in all 20 cases had the least significant total changes in gene expression. This unexpected finding was true for each pathotype and between each of the three assessed donors. It is tempting to speculate that the reverse is true here; the host purposely suppresses inflammation in this region. It is seemingly a paradox that this would occur in the segment of the GI tract with the most gut associated lymphoid tissue (GALT). One hypothesis to explain this finding is that the epithelial cells in the ileum are purposefully not responding to enteric pathogens so as to not create a local vicious cycle of unregulated stimuli/responses that might damage the ability to coordinate a specific immune response; it could also be that the responsiveness of the ileum to infection occurs only after receiving paracrine signals created by other GI segments or from Microfold cells (M cells). More work will be needed to test these ideas.

A third finding is that the genes that are differentially regulated are generally not shared between each segment, comprising, on average, only a few percent of the total response (with the exception of the fimbrial mutant of EAEC, see below), despite the fact that the duodenum, ileum, and colon segments assessed here came from the same donor. This finding suggests we must consider each segment of the intestine as a distinct and separate tissue that undergoes its own tailored host response to each different pathotype. With the growing emergence of the concept of "biogeography", these findings are especially important. Fourth, wide-scale host responses are observed even in the absence of adherence. For example, the jejunum does not support strong aggregation or total bacterial binding of EAEC [35], yet there were hundreds of genes either upregulated or downregulated in this tissue.

Finally, there were literally thousands of genes that were differentially up or down in each segment and amongst the different pathotypes (Appendix 2 lists all of them, rank-ordered). This treasure trove of information should continuously be mined by various researchers to identify patterns that might help us predict susceptibility and disease severity. Notable changes

for EAEC include genes such as: IL-8, IL-1β and IL-1A. IL-8 in particular has allelic variance in the human population and this correlates with infection [9, 31, 37]. There were other pathways as well, including strong positive upregulation of cellular proliferation and apoptosis pathways, which may be a response to heal the epithelium. One of the more striking findings though was the up regulation of intestinal mucins. We interpret this data to suggest that these mucins, and mucus in general, are as important as any response in the host attempting to strengthen barrier integrity and to generate an interface that separates the host from pathogen. The finding, however, that EAEC binds HSPGs, a structure that is also enriched on mucin proteins, is a testament to the remarkable adaptability of pathogenic bacteria, and *E. coli* more specifically. In this regard, one wonders if such a host response is a double-edged sword; on the one hand the host production of sugar decoys may increase bacterial shedding and keep bacteria away from such receptors on the host epithelial cells. On the other hand, such ligands may provide additional handles for the bacterium to further colonize. Future work should attempt to address these possibilities, as well as those speculated above.

## Materials and methods

### Bacterial strains

The bacterial strains used were: EAEC 042 (serotype 044:H18) [39], EAEC 042: Δ*aafA*, EAEC, *E. coli*: HS, EHEC 8624 [40], ExPEC CP9 (serotype O4:K53:H5) [41, 42] and EAEC A2A (a clinical isolate). The list and the sources of all strains used are provided in Table 1. Bacteria were grown in Tryptic Soy Broth at 37˚C overnight and sub-cultured for at least 2 hours before every infection. All infections are performed at a multiplicity of infection (MOI) of 10 and a total infection time of 3 hours unless otherwise indicated.

### Generation of EAEC: GFP

EAEC A2A strain was genetically modified by insertion of a chloramphenicol resistance and superfolder green fluorescent protein (sfGFP) gene by classical P1 phage transduction [96]. The construct was generated by the laboratory of Dr. Christophe Herman originally in E. coli K12 (MG1655). Modifications (sfGFP:Cat:FRT) were transformed into EAEC using P1 phage and integrated at the nfsA locus. Transformants were selected on Chloramphenicol plates. Selected transformants were assessed for fluorescence and confirmed using oligos for upstream and downstream of the nfsA site (FWD: GGTGGTTATTCTTCAGGTG and REV: CGCTCCAGGTATCCC).

**Table 1. Details of *E. coli* strains used in the study.**

| No. | Strains | Characteristics | Source |
|---|---|---|---|
| 1 | EAEC 042 | Wild-type and prototype strain | PCO |
| 2 | EAEC A2A | Clinical isolate | PCO |
| 3 | *E. coli* HS | Lab adapted commensal | PCO |
| 4 | EAEC042: ΔaafA | aafA, a major fimbrial subunit mutant | JN |
| 5 | EHEC 8624 | Wild-type strain | AT |
| 6 | ExPEC CP9 | Wild-type strain | JJ |

PCO- Dr. Pablo C. Okhuysen (UT Health, MD Anderson Cancer Institute), JN- Dr. James Nataro (University of Virginia School of Medicine), AT- Dr. Alfredo Torres (University of Texas Medical Branch), JJ- Dr. James Johnson (University of Minnesota)

### Ethics statement

All human sample/tissue collection was performed in accordance with the Baylor College of Medicine Institutional Review Board regulations and were collected under protocol H-35094 with informed consent from all adult subjects participating in the study.

### Human intestinal enteroid culture

Human Intestinal Enteroid Monolayers (HIEMs) were derived and cultured from 3D enteroid cultures of D103,D104, D109, J2, J3, J11, I103, I104, I109, TC202, C103, C104 and C109, as previously described [35, 97]. The apical-out colonoids cultures were generated as previously described [46].

### Global gene expression

Infections were performed on enteroid monolayers and lines as described above after being differentiated for five days. After infection, host cells were lysed using Trizol and RNA extracted using previously published protocols [56].

### Library preparation, construction and sequencing

mRNA was enriched using oligo (dT) beads, fragmented randomly in fragmentation buffer, followed by cDNA synthesis using random hexamers and reverse transcriptase. After first-strand synthesis, a custom second-strand synthesis buffer (Illumina) was added with dNTPs, RNase H and *Escherichia coli* polymerase I to generate the second strand by nick-translation. The final cDNA library was ready after a round of purification, terminal repair, A-tailing, ligation of sequencing adapters, size selection and PCR enrichment. All processing steps were performed by Novogene, USA.

### RNA-Sequencing analysis, quality control, and results

Fastq files were first trimmed to remove adapters and base-pairs that did not meet quality standards using Trim Galore (Version 0.4.1) [98, 99]. Trimmed sequences are aligned against the human genome hg38 (GRCh38) genome using Hisat2 (version 2.1.0) and sorted using samtools (version 1.5) [100, 101]. Next, a count matrix was generated by mapping reads using featureCounts (version 1.6.0) [102], and only non-overlapping uniquely mapped features were counted.

### Expression quantification and gene set enrichment analysis

Differential gene expression was determined using the R-packages edgeR, limma and voom following published guidelines [103, 104]. Briefly, the feature count matrix was filtered for the coding genes only, and then a cutoff of 1 count per million was used to remove genes with low expression. The filtered counts are normalized using the trimmed mean of M-values (TMM) approach. The R-package voom was then used to remove the dependence of the variance on the mean gene expression, before determining differential gene expression using the limma R-package. The Benjamini-Hochberg (BH) correction was applied to the p-values to account for multiple hypothesis testing (adjusted p-values). Genes are ranked by their log2 fold gene expression for use in gene set enrichment analysis (GSEA version 3.0). GSEA was performed using these ranked gene lists against the entire MSigDB database (version 6.1) [103, 104].

## Principal component analysis (PCA)

PCA is a procedure that uses orthogonal transformations to reduce the number of variables. The principal components are sorted by decreasing amounts of explained variability in the data. Plotting the transcriptome profiles based on the first three principal components is used to reveal relationship between samples. Hierarchical clustering of samples was performed using the Pearson correlation as distance metric, based on used log2CPM transformed and filtered data, after correction for batch and patient variation.

## Preparation of coverslips for HIE monolayer culture for SEM

12 mm diameters round #1.5 coverslips were modified with 3-(trimethoxysilyl) propyl methacrylate (TMSPMA) to create a surface favorable for collagen IV adsorption. To clean and activate the surface of the coverslips, they are treated with piranha solution (3 parts sulfuric acid to 1-part hydrogen peroxide) for 1 hour. After washing with distilled water and 95% ethanol, the coverslips are incubated overnight at room temperature with a 2% solution of TMSPMA in 95% ethanol pH 4.5. Next, they are washed with 95% ethanol, dried with a laboratory wipe, and baked at 90˚C for 1 hour. To culture HIEs on top of the modified coverslips, temporary wells are formed from 3 mm thick sheets of silicone rubber. Circular wells with a 6 mm inner diameter and 10 mm outer diameter are created with hand punches. These wells are sterilized by autoclaving and applied to the coverslips with tweezers to form a watertight seal. A dilute collagen IV solution was added to these wells for 90 minutes at 37˚C, and then HIEs are seeded into them.

## Scanning electron microscopy (SEM) and transmission electron microscopy

HIEMs are plated either on TMSPMA coated 12 mm diameters round #1.5 coverslips (Fisher scientific, Cat # NC1129240) or 96-well Nunc cell culture plates (Corning, Cat# CLS3595) for SEM and TEM imaging respectively. After 4 days of differentiation, HIEMs are infected with EAEC A2A at a MOI of 10 for 3 hours. For SEM, after post-infection, HIEMs are fixed with glutaraldehyde and dehydrated gradually in a graded series of ETOH- 30%, 50%, 70%, 95%, and 2x 100% for 15–30 minutes. The specimens are then progressively substituted for ethanol, followed by complete evaporation of ethanol and coated with gold to be observed under SEM. For TEM, after infection, HIEMs are trypsinized, pelleted in PBS and fixed with glutaraldehyde and dehydrated in graded ethanol. The specimens are then progressively substituted for ethanol, followed by complete evaporation of ethanol and coated with gold to be observed under TEM.

## Histology staining and Periodic Acid Schiff/ Alcian Blue staining

HIEMs cultured on Transwell and at 4 days of differentiation are infected with EAEC at a MOI of 10 and incubated for 3 hours at 37˚C in the presence of 5% $CO_2$ in a humidified incubator. The cells are washed three times with PBS gently to remove any non-adherent bacteria. The cells are fixed with 4% paraformaldehyde (PFA), progressively dehydrated in ethanol in a graded series of 30%, 50%, 70%, 95%, and 100% for 15–30 minutes at room temperature, embedded in paraffin and serially sectioned to 5 µm sections. Periodic acid Schiff (PAS) and Alcian blue (AB, pH2.5) was used to investigate the presence of goblet cell secreted mucins in EAEC clusters. All goblet cell mucus are labeled in light blue, nuclei in red to pink and cytoplasm in pale pink. Paraffin embedded sections are deparaffinized, oxidized in periodic acid and rinsed in water. Later they are then stained with Schiff's reagent, rinsed with water,

followed by 1% Alcian Blue (AB) staining in 3% acetic acid at a pH 2.5 and rinsed with water. Sections are counterstained with hematoxylin, washed, dehydrated and mounted for visualization under light microscopy.

### Immunofluorescence staining

To identify association of EAEC aggregative adherence to MUC2, GFP-labelled EAEC: A2A was used. HIEMs plated on 96 well iBidi μ plates were fixed with Clarks fixative and incubated for 10 minutes [45]. HIEMs were permeabilized and blocked with 5% bovine serum albumin (BSA) in 0.1% Triton X-100 in PBS for 30 min at RT. MUC2 and cell boundaries were detected using the following antibodies: MUC2 (1:200) (Abcam: ab 11197) and Alexa Fluor E-cadherin (1:200) (BD Bioscience 560062) after overnight incubation at 4˚C. Nuclei were stained with 4′, 6′-diamidino-2-phenylindole (DAPI) (300 nM) for 5 minutes at RT. Orthogonal sections of the sample were captured using a Zeiss LSM 510 confocal microscope.

### Giemsa wright staining

HIEMs cultured on chambered slides and at 4 days of differentiation were infected at a MOI of 10 and incubated for 3 hours at 37˚C in the presence of 5% $CO_2$ in a humidified incubator. The cells were washed three times with PBS to remove any non-adherent bacteria. The cells were then fixed and stained using Hema fixative and solutions (Protocol, Cat#122–91) and imaged under oil immersion microscopy at 100x.

### AafA expression and purification

AafA is the major subunit of the AAF/II pilus which is assembled by the FGL (long looped) chaperone/usher mechanism [105, 106]. During pilus polymerization, the flexible N-terminal extension (donor strand) of AafA binds to the hydrophobic groove of the adjacent subunit to complete the Ig-like fold, referred to as donor-strand exchange [107–110]. To generate a donor-strand complemented AafA expressed in the cytoplasm, the insert consisted of a 2-residue linker followed by residues 11–135 at the N-terminus, a ten-residue linker, and residues 1–10 at the C-terminus ligated into vector pQE-30 (Qiagen) using restriction sites BamHI and HindIII. The AafA sequence [108] was obtained from (GenBank: AAB82330.1), and the signal peptide was removed to express and retain the protein in the cytoplasm. Constructs with this rearrangement of residues and the incorporated linker allow for conformational flexibility between the N and C-terminus and have been shown to produce soluble and stable donor strand complemented AafA monomers [107–110]. To separate the His6 tag and also facilitate the flexibility, a short Gly-Ser linker was also incorporated at the N-terminus (atg cat cac cat cac cat cac ggt agt aat ttt tgt gat ata acg ata aca ccg gct aca aat cgt gat gtc aac gtt gac agg agc gca aat atc gac ctg agt ttt act att aga caa ccg caa cgc tgc gct gat gct ggt atg cga ata aaa gct tgg ggg gaa gcc aat cac ggt caa tta ctg ata aaa cct caa gga gga aat aaa tca gca gga ttc act ctg gcc tct cct agg ttt tct tac att ccg aat aat cca gca aac att atg aat gga ttt gtt ctt acg aat cct ggt gtt tat caa tta gga atg cag ggc tca att aca ccg gct atc ccg cta cga cca ggc cta tat gaa gta gta tta aat gct gag ctt gtg aca aat gat aac aag caa aat gca act gcg gta gca aaa act gcg acc agt act atc act gta gtg taa)His6-Gly-Ser-AafA was expressed in XL1-Blue Supercompetent Cells (New England Biolabs) which allows tight regulation of potentially toxic genes by laclq. Cells were induced with 1mM IPTG at an OD600 of 0.6 for 4 hours, harvested by centrifugation at 4000 x g for 20 minutes, and the pellet was stored overnight at -20˚C. The next day the pellet was suspended in lysis buffer (100 mM NaH2PO4 10 mM Tris·Cl 8 M urea, pH 8.0) and French pressed. AafA was purified by Ni-NTA (Qiagen) according to manufacturer's instructions.

## Glycan microarray preparation and analysis

Oligosaccharide array composed of 560 defined glycans from Consortium for Functional Glycomics were from MicroArray Core Facility of the Scripps Research Institute (La Jolla, CA) [47]. Glycans were printed on Nexterion 3-D hydrogel coating (H) slides using a contact printer in replicates of six described previously (1). A second microarray consisting of 170 charged glycosaminoglycans (GAG), including oligomers from hyaluronic acid, chondroitin sulfates and heparan, were from Emory Comprehensive Glycomics Core (Atlanta, GA). Glycans were conjugated with bifunctional linker AEAB and printed on Nexterion 3-D hydrogel coating (H) slides using a contact printer Aushon 2470 (Quanterix, Billerica, MA) in replicates of four. His-tagged AafA was diluted in PBS buffer with 1% BSA and 0.05% Tween 20 and incubated on the microarray slides for 1 hour at room temperature. Slides were washed with PBS buffer with 0.05% Tween 20 and PBS buffer. Alexa Flour 647 anti-His-tag mouse mAb (MBL International, Woburn, MA) was diluted to 5 μg/ml and incubated on the slides for 1 hour. After washing and drying, slides were scanned with InnoScan 1100 AL scanner and data were processed using Mapix 8.2.5 software (Innopsys, Chicago, IL). Mean Relative Fluorescence Units (RFU) from replicate spots after subtracting local median background were averaged, and standard deviation (STDEV) were calculated and plotted as Error Bar in the histogram plot of glycan array result.

## CFU counting

After infection, the plate was tipped over for gentle removal of the media (to prevent removal of the mucus layer). Cells were washed gently with RT PBS 3x to remove un-adhered bacteria and each time; the plate was tipped to remove the PBS. 100ul cold PBS was added to each well and cells were scraped off with a P1000 tip to recover adhered bacteria. This solution was serially diluted to $10^{-7}$ in PBS and plated on LB agar. Colonies were counted the following day for each dilution and CFUs per monolayer calculated.

## Statistical analysis

A statistical significance was determined by two-way ANOVA. Tukey's multiple comparison tests are performed using GraphPad Prism version 7.0 Windows (Graph Pad Software, San Diego, CA, www.graphpad.com). Differences in the means are considered significant at *$p \leq 0.05$ with specific p values detailed in the figure legends. The model figures were created with BioRender.com.

## Funding information

These studies were supported in part by grant U19 AI11497 that is as part of the U19 program (Novel Alternative Models of Enteric Diseases–NAMSED) from the National Institutes of Health assigned to MKE and AWM and from seed funds NAMSED institutional 1383006110 to AWM from Baylor College of Medicine. This study was partially supported by NIH P30 shared resource grant CA125123, NIEHS grants 1P30ES030285 and 1P42ES0327725 (MJR,CC) and by The Cancer Prevention Institute of Texas (CPRIT) RP170005. The funders had no role in study design, data collection and analysis, decision to publish, or preparation of the manuscript. The authors sincerely apologize for any work left uncited do to journal space constraints.

## Supporting information

**S1 Fig.** (A) Principal component analysis of jejunum and ileal monolayers demonstrating the variability observed in the samples (infected and mock controls). (B) Total number of

significant genes identified for HS, ΔaafA, EHEC and ExPEC-infected groups (BH adjusted p-value < 0.05).
(TIF)

**S2 Fig. Volcano plot of differential gene expression in each tissue for the HS, ΔaafA, EHEC and ExPEC-infected samples compared to the mock-infected controls.** Red dots indicate genes that have an adjusted p-value of at least 0.05 with a linear fold change of at least 1.5.
(TIF)

**S3 Fig. Venn diagram representing the number of genes that are uniquely expressed: duodenum, jejunum, ileum and colon of HS, ΔaafA, EHEC and ExPEC-infected samples (p-value < = 0.05).**
(TIF)

**S4 Fig. Heatmap for HS, ΔaafA, EHEC and ExPEC-infected samples for each intestine section (Ebayes test, adjusted p-value < = 0.05 and Fold Change = 1.5).** The number of significant genes is indicated on the right of each individual heatmap.
(TIF)

**S5 Fig. Select GO pathways identified from a GSEA analysis by filtering for the top ten Normalized Enrichment Score (NES) for each category.** False discovery rate (FDR) of 0.05 was used in the pathway filtering.
(TIF)

**S6 Fig.** (A) Coomassie blue stained SDS-PAGE of AafA. Molecular weight ladder (MW), Flow through (FT), Washes 1 and 2 (W1-2), and Elutes 1–6 (E1-6) of AafA-His protein at 18kDa after purification by Ni-NTA chromatography. (B-C) Binding pattern of AafA at 2µg/ml and 50µg/ml on a pan glycan array (composed of >500 synthetic glycan) shows preferential binding to charged glycans. (D) Quantification of the total level of EAEC adherence to HIEMs. EAEC or HSS-EAEC were incubated with 2D differentiated colon 109 monolayers that were either mock treated or with Heparinase III and adherence quantified as described in the Methods.
(TIF)

**S7 Fig. Model for EAEC adherence on human intestinal enteroids.**
(TIF)

**S1 Appendix. Set of Up-regulated and Down-regulated genes in HIEs.** The excel sheets lists all the different genes that are up- or down-regulated in HIEs to bacterial infection across four segments of intestine and from three patients.
(XLSX)

**S2 Appendix. Differentially Expressed Gene Signature in HIEs.**
(XLSX)

**S3 Appendix. List of all GSEA pathways.**
(XLSX)

**S4 Appendix. Results from pan and GAG-specific glycan array.**
(XLSX)

**S1 Video file. Real-time growth of EAEC, EAEC:ΔaafA and HS in HIEs.**
(MP4)

## Author Contributions

**Conceptualization:** Anubama Rajan, Anthony W. Maresso.

**Data curation:** Anubama Rajan, Matthew J. Robertson, Cristian Coarfa, Anthony W. Maresso.

**Formal analysis:** Anubama Rajan, Matthew J. Robertson.

**Funding acquisition:** Cristian Coarfa, Jane Grande-Allen, Mary K. Estes, Pablo C. Okhuysen, Anthony W. Maresso.

**Investigation:** Anubama Rajan, Anthony W. Maresso.

**Methodology:** Anubama Rajan, Hannah E. Carter, Nina M. Poole, Justin R. Clark, Sabrina I. Green, Boyang Zhao, Yikun Xing, Reid L. Wilson, Fan Bai.

**Project administration:** Anthony W. Maresso.

**Resources:** Joseph M. Hyser, Joseph Petrosino, Noah F. Shroyer, Sarah E. Blutt, Xuezheng Song, BV Venkataram Prasad, Mary K. Estes, Anthony W. Maresso.

**Software:** Anubama Rajan, Matthew J. Robertson, Justin R. Clark.

**Supervision:** Anthony W. Maresso.

**Validation:** Anubama Rajan, Hannah E. Carter, Zachary K. Criss, Nikhil Jain.

**Visualization:** Anubama Rajan, Umesh Karandikar, Mar Margalef-Català, Manuel R. Amieva.

**Writing – original draft:** Anubama Rajan.

**Writing – review & editing:** Anubama Rajan, Anthony W. Maresso.

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
