## [Decision Letter · Decision Letter 0]

25 Jun 2020

Dear Dr. Maresso, 

Thank you very much for submitting your manuscript "Enteroaggregative E. coli Adherence to Human Heparan Sulfate Proteoglycans Drives Segment and Host Specific Responses to Infection" (PPATHOGENS-D-20-01075) for review by PLOS Pathogens. Your manuscript was fully evaluated at the editorial level and by 3 independent peer reviewers. The reviewers appreciated the attention to an important topic but identified some aspects of the manuscript that should be improved. We therefore ask you to modify the manuscript according to the review recommendations before we can consider your manuscript for acceptance. Your revisions should address the specific points made by each reviewer.

In particular you should note that all three reviewers felt that the manuscript was of importance, but the first reviewer had some concerns that would require additional experimentation. In particular, this reviewer requested that you consider complementation of the aafA-mutant of EAEC strain 042. I agree that to completely fulfill "molecular Koch's postulates" that this complementation is required. In you earlier mBio report, you also did not use a complemented strain. In that report I did not find any reason to why complementation was not attempted. As is true for many strains of E. coli, including clinical isolates, there is sometimes difficulty encountered when trying to perform multiple genetic manipulations as opposed the case with "domesticated" laboratory strains. Has complementation been attempted for this strain?

(As a side note, I would ask that you check your references. For example in the Material and Methods, reference to the EAEC strain 042 is reference "99". This paper does not refer to this strain by my reading but is a instead a paper concerning EAF in EPEC strain E2348/69. Similarly in the text [line 129] you refer to EHEC strain 86 [which I believe is more completely referred to as "86-24" via reference 39. This reference does not use 86-24, but does use EDL933 as a "prototypic" EHEC strain. I did not exhaustively go through your references, but since I found these two when I looked, I would ask that you carefully look through the entire reference set. You should note that reviewer 3 also had some questions about the sources of your strains.)

If complementation has been attempted, but not successful, this could be noted. While these data would be useful if available, I am willing to forego this suggestion if you present rationale for not complementing the aafA mutant.

Another area of specific concern raised by both reviewers 1 and 2 concerns the cloning and expression of aafA. My read is that you expressed the entire protein, without the leader peptide. Clarification would be helpful and also addressing the need for the linker region you added.

Of course, a point-by-point of the full set of concerns from the 3 reviewers will be required, but I wanted to highlight what I felt were major issues, especially regarding the need for additional experimentation raised by the first reviewer.

Sincerely,

Vincent B. Young

Guest Editor

PLOS Pathogens

Guy Tran Van Nhieu

Section Editor

PLOS Pathogens

Kasturi Haldar

Editor-in-Chief

PLOS Pathogens

orcid.org/0000-0001-5065-158X

Michael Malim

Editor-in-Chief

PLOS Pathogens

orcid.org/0000-0002-7699-2064

Dear Dr. Maresso,

Thank you very much for submitting your manuscript "Enteroaggregative E. coli Adherence to Human Heparan Sulfate Proteoglycans Drives Segment and Host Specific Responses to Infection" (PPATHOGENS-D-20-01075) for review by PLOS Pathogens. Your manuscript was fully evaluated at the editorial level and by 3 independent peer reviewers. The reviewers appreciated the attention to an important topic but identified some aspects of the manuscript that should be improved. We therefore ask you to modify the manuscript according to the review recommendations before we can consider your manuscript for acceptance. Your revisions should address the specific points made by each reviewer.

In particular you should note that all three reviewers felt that the manuscript was of importance, but the first reviewer had some concerns that would require additional experimentation. In particular, this reviewer requested that you consider complementation of the aafA-mutant of EAEC strain 042. I agree that to completely fulfill "molecular Koch's postulates" that this complementation is required. In you earlier mBio report, you also did not use a complemented strain. In that report I did not find any reason to why complementation was not attempted. As is true for many strains of E. coli, including clinical isolates, there is sometimes difficulty encountered when trying to perform multiple genetic manipulations as opposed the case with "domesticated" laboratory strains. Has complementation been attempted for this strain?

(As a side note, I would ask that you check your references. For example in the Material and Methods, reference to the EAEC strain 042 is reference "99". This paper does not refer to this strain by my reading but is a instead a paper concerning EAF in EPEC strain E2348/69. Similarly in the text [line 129] you refer to EHEC strain 86 [which I believe is more completely referred to as "86-24" via reference 39. This reference does not use 86-24, but does use EDL933 as a "prototypic" EHEC strain. I did not exhaustively go through your references, but since I found these two when I looked, I would ask that you carefully look through the entire reference set. You should note that reviewer 3 also had some questions about the sources of your strains.)

If complementation has been attempted, but not successful, this could be noted. While these data would be useful if available, I am willing to forego this suggestion if you present rationale for not complementing the aafA mutant.

Another area of specific concern raised by both reviewers 1 and 2 concerns the cloning and expression of aafA. My read is that you expressed the entire protein, without the leader peptide. Clarification would be helpful and also addressing the need for the linker region you added.

Of course, a point-by-point of the full set of concerns from the 3 reviewers will be required, but I wanted to highlight what I felt were major issues, especially regarding the need for additional experimentation raised by the first reviewer.

Reviewer Comments (if any, and for reference):

Reviewer's Responses to Questions

**Part I - Summary**

Reviewer #1: Manuscript control number: PPATHOGENS-D-20-01075

Enteroaggregative E. coli Adherence to Human Heparan Sulfate Proteoglycans Drives Segment and Host Specific Responses to Infection.

Rajan A., et al.

This manuscript evaluates the interactions between enteroaggregative Escherichia coli (EAEC) and host intestinal cells and their impact on tissue tropism and host cells responses. They showed that the host responses varied and are linked to either the pathogen colonizing the specific site or the host-associated responses. An interesting finding is the association between EAEC and intestinal mucins, a process that requires a functional AafA fimbriae. Finally, the authors demonstrated that AafA binds to heparan sulfate proteoglycans. Although the paper goal is to demonstrate that bacterial AafA-mediated interactions are responsible for the association with tropism and mucin interactions, the specific contribution of this fimbriae needs further elucidation. Therefore, further experiments and revisions of the paper are required and summarized next for authors’ consideration:

Reviewer #2: This manuscript contains a very large amount of information. It provides the initial analysis of the response of HIEs to different pathotypes of E. coli, and it further shows the responses for different patients and different intestinal segments. These observations are important and will guide future work in the field. The interaction between EAEC and the human tissues is explored in more detail, including analysis of the role of the adhesin AafA and host heparan sulfate proteoglycans in the interactions. While there are a number of additional experiments that come to mind, those likely will be the subjects of future work. The current manuscript includes a wealth of data and several clear conclusions. Thus, my suggestions are largely cosmetic or clarifications.

Reviewer #3: A major problem for understanding infectious intestinal disease has been that mice are not susceptible to many human intestinal pathogens. Furthermore, human cell lines cannot model complex tissue responses. This study uses multi-potent stem-cell derived human intestinal tissues from three individuals and four different area of the human intestinal tract to study commensal E. coli and different pathovars. They generate, for the first time, a highly detailed analysis of human-specific responses to human intestinal pathogens. Most of the information comes from using RNA-seq and confocal microscopy. The authors examine both host and pathogen factors, and find complex responses, that are sometime tissue dependent and sometimes pathogen dependent. The data are very robust, integrating state-of-the-art techniques, leading to well-supported paradigm-shifting conclusions. The manuscript is extremely well-written, and only a few areas could be clarified as detailed below.

**Part II – Major Issues: Key Experiments Required for Acceptance**

Reviewer #1: Major points:

1. Lane 128 and Fig 1. Strain 86 is an EHEC non-Stx- producing strain, so the authors might want to clarify the reason of using such stx-negative strain for their experiments.

2. Lanes 161-165, a significant concern regarding the interpretation of the results is the use of an aafA-negative, which is expected to have a reduced adherence and therefore, a reduced impact in the total gene expression changes is different intestinal segments. This observation raises multiple questions that require attention in this manuscript: a) does the aafA-mutant strain uses another mechanism to interact with the host cells? b) a complementation experiment is required to confirm that the effect is directly associated with AafA and no other non-specific mutations.

3. Lanes 190-192, it is not clear to the reader the reason the authors consider ExPEC the “most pathogenic” pathotype and why is more like the aafA mutant strain. The data might suggest then that the aafA-mutant strain modifies its tropism but maintains its ability to bind to other intestinal segments, causing further damage mediated by other adhesins? If that is the case, the authors need to confirm this finding or confirm the specificity of AafA tissue interaction.

4. Lanes 203-205. As indicated above, the drastic changes observed with the aafA mutant strain require a complementation analysis to confirm that this is an AafA-specific phenotype and not an artifact of the mutation, because it is difficult to understand how a reduction in adherence will have such a drastic change in host-responses.

5. Lane 244 “…different adherence patterns on enteroids” have not been shown in this manuscript. Is this part of a prior publication? Otherwise an example adherence patterns must be included.

6. Lanes 266-270, it is not clear the rationale behind the selection of MUC2 for the staining in Fig 4, because MUC2 was not one of the mucins differentially expressed upon contact with EAEC. It will make more sense to use a MUC that is associated with EAEC interaction.

7. Lanes 305-306, again, it is not clear the reason to incorporate a 10-residue linker to the segment of the aafA gene cloned? Is the gene segment sufficient to express the functional binding domain?

8. Discussion, this is an extremely lengthy discussion and at least it must be reduced by 50%. For example, lanes 344-355 includes an introduction rather than a discussion, particularly since the organotypic model might not be useful to evaluate a vaccine.

Reviewer #2: (No Response)

Reviewer #3: none

**Part III – Minor Issues: Editorial and Data Presentation Modifications**

Reviewer #1: Minor points:

1. Lane 92 “…ecological niche; second, what are…”

2. Lane 96, do you mean “ex vivo” instead of “ex situ”?

3. Lane 113 and elsewhere. The protein name is “AafA” and the gene name “aafA”

4. Lanes 120-123, this is a run-on sentence, please re-structure.

5. Figure 2, order of panels, panels D and E should be C and D and then panel C should be E. change in the text.

6. Lanes 254-260 and Figure 3. It is not clear which ones are panels 1-4, do you refer to the rows in panels A-C?

7. Lane 289 and Fig 5D, it should read “10 µM”

8. References, italicize “Escherichia coli”, “in vitro“, “E. coli“, “in vivo”, “Campylobacter jejuni”, “Staphylococcus aureus”. Reference 33 remove all capital letters. Reference 35 “mBio” instead of “MBio”.

Reviewer #2: 1. In the comparison of the WT and aafA mutant, it is shown that the mutant induced a much broader host response than the wild type. Is it possible that the fimbrae may have a role in suppressing host response? Were cytokine or innate immune response genes among those that were upregulated in the presence of the mutant but not the wild type?

2. Line 304. …amino acids 1-135 of AafA. Is this the full length protein? If so, it would be clearer to say the full length protein was cloned and put the details in the Materials and Methods. If this is a fragment of the protein, some explanation for the fragment chosen would be useful.

3. Fig. 3. Some explanation is needed for what is counted as a cluster. It is not clear from the legend what the bars in Fig. 3D mean. Also, the legend says the yellow arrow indicates EAEC clusters but these look more like individual bacteria in the 4th panel.

4. Fig. 4 and Fig. 5 legends. DAPI (nuclei) and GFP (EAEC) are defined but the staining for MUC2 and E-cadherin are not stated.

5. Fig. 5 D and E. Is this the number of aggregates per field, per sample?

6. Fig. 6. A bit more detail is needed in the legend for D-F. Defining the acronyms and stating that soluble HS is added as a competitor would make this easier for the reader.

Reviewer #3: For experimental clarity:

This ground-breaking study is likely to drive many other investigations in the field. As such, it would be useful if the methods were very clear for others to follow. Something akin to the Key Resource Table introduced by CellPress should be adopted by all journals.

1. In general, the figure legends lack detail and should include number of repeats and the statistical analyses used. For example: Figure 2E, 3D, 5D-E. What is N? These look like pairwise comparisons, was Tukey’s multiple comparison really used for the statistical analysis?

2. The Materials and Methods also lack detail, particularly the sections written in present tense. They should include the source of reagents. Including references to published protocols would be helpful. Some specific questions:

Line 641 – What are 10¢ coverslips

Line 644 – describe graded ethanol

Line 654 – clarify or reference progressively dehydrated in ethanol

Line 657- 658, please clarify “acidic mucus substances are labeled in blue, neutral mucins in purple, nuclei in red to pink and cytoplasm in pale pink”.

3. line 313 – why is 2,000 considered background?

4. Lines 684-694 - A BLAST search revealed 4 mismatches at the sequence beginning to the best hit in GenBank. Please indicate the source of the gene information.

5. Strains – please reference the source of EAEC 042: ΔAafA, E. coli: HS and describe the construction of EHEC-GFP. A table would be helpful.

Minor typos and suggestions:

1. Line 277, should be Co, et al.

2. Line 289 – symbol error

3. Figure 2A, the second and third panels have been cut off by an extra data label.

4. Figure 2B, legend on right, the data order, listed up to down, is reversed from the left to right data order in the figure.

5. Sometimes elevated expression is red (or yellow), sometimes it is blue. It would be best to be consistent.

6. Figure 3A-C, For clarity, can you change “panel one” to “row 1”, etc. ?

7. Figure 4. Which coronoid line or lines were used?

8. Figure 4B. The insert needs more explanation. What cells are purple? Are the goblet cells in green? This is confusing since in the accompanying image the bacteria are stained in green.

9. Figure 6B. Please define the symbols in panel B.

10. Figure 6C. Where is R84?

11. Line 1097 – define FC

12. Line 1167-1168. Change “bottom” to “on the right”.

13. Line 1171 – define NES, FDR

14. Supporting information – please make sure all data entries are define – for example, D,J,I,C in: D.HS.over.Mock, J.HS.over.Mock, I.HS.over.Mock, C.HS.over.Mock

PLOS authors have the option to publish the peer review history of their article (what does this mean?). If published, this will include your full peer review and any attached files.

Reviewer #1: No

Reviewer #2: **Yes: **Shelley M. Payne

Reviewer #3: No
---

## [Editor Report · Decision Letter 1]

1 Aug 2020

Dear Dr. Maresso,

We are pleased to inform you that your manuscript 'Enteroaggregative E. coli Adherence to Human Heparan Sulfate Proteoglycans Drives Segment and Host Specific Responses to Infection' has been provisionally accepted for publication in PLOS Pathogens.

Best regards,

Vincent B. Young

Guest Editor

PLOS Pathogens

Guy Tran Van Nhieu

Section Editor

PLOS Pathogens

Kasturi Haldar

Editor-in-Chief

PLOS Pathogens

orcid.org/0000-0001-5065-158X

Michael Malim

Editor-in-Chief

PLOS Pathogens

orcid.org/0000-0002-7699-2064
---

## [Editor Report · Acceptance letter]

11 Sep 2020

Dear Dr. Maresso,

We are delighted to inform you that your manuscript, "*Enteroaggregative E. coli* Adherence to Human Heparan Sulfate Proteoglycans Drives Segment and Host Specific Responses to Infection," has been formally accepted for publication in PLOS Pathogens.

Best regards,

Kasturi Haldar

Editor-in-Chief

PLOS Pathogens

orcid.org/0000-0001-5065-158X

Michael Malim

Editor-in-Chief

PLOS Pathogens

orcid.org/0000-0002-7699-2064